# Recent Advances in the Immunologic Method Applied to Tick-Borne Diseases in Brazil

**DOI:** 10.3390/pathogens11080870

**Published:** 2022-08-02

**Authors:** Mônica E. T. Alcon-Chino, Salvatore G. De-Simone

**Affiliations:** 1Center for Technological Development in Health (CDTS), National Institute of Science and Technology for Innovation in Neglected Population Diseases (INCT-IDPN), FIOCRUZ, Rio de Janeiro 21040-900, Brazil; elizaalcon@gmail.com; 2Post-Graduation Program in Science and Biotechnology, Department of Molecular and Cellular Biology, Biology Institute, Federal Fluminense University, Niterói 22040-036, Brazil; 3Laboratory of Epidemiology and Molecular Systematics, Oswaldo Cruz Institute, FIOCRUZ, Rio de Janeiro 21040-900, Brazil

**Keywords:** tick-borne diseases, ehrlichiosis, borreliosis, Brazilian Spotted Fever, Lyme disease, immunologic diagnosis, serological diagnosis, biosensors

## Abstract

Zoonotic-origin infectious diseases are one of the major concerns of human and veterinary health systems. Ticks, as vectors of several zoonotic diseases, are ranked second only to mosquitoes as vectors. Many ticks’ transmitted infections are still endemic in the Americas, Europe, and Africa and represent approximately 17% of their infectious diseases population. Although our scientific capacity to identify and diagnose diseases is increasing, it remains a challenge in the case of tick-borne conditions. For example, in 2017, 160 cases of the Brazilian Spotted Fever (BSF, a tick-borne illness) were confirmed, alarming the notifiable diseases information system. Conversely, Brazilian borreliosis and ehrlichiosis do not require notification. Still, an increasing number of cases in humans and dogs have been reported in southeast and northeastern Brazil. Immunological methods applied to human and dog tick-borne diseases (TBD) show low sensitivity and specificity, cross-reactions, and false IgM positivity. Thus, the diagnosis and management of TBD are hampered by the personal tools and indirect markers used. Therefore, specific and rapid methods urgently need to be developed to diagnose the various types of tick-borne bacterial diseases. This review presents a brief historical perspective on the evolution of serological assays and recent advances in diagnostic tests for TBD (ehrlichiosis, BSF, and borreliosis) in humans and dogs, mainly applied in Brazil. Additionally, this review covers the emerging technologies available in diagnosing TBD, including biosensors, and discusses their potential for future use as gold standards in diagnosing these diseases.

## 1. Introduction

The arthropod vectors include many tiny organisms such as mosquitoes, sandflies, aquatic snails, blackflies, triatome, lice, tsetse, tick, and fleas, transmitting pathogens [1]. Vector-borne diseases (VBDs) significantly impact human health and veterinary health systems [2]. In 2020, the World Health Organization [WHO] reported that around 17% of infectious diseases causing more than 700,000 deaths are related to VBDs [1].

Ticks, as vectors of several zoonotic diseases, are ranked second only to mosquitoes as vectors. Among tick-borne conditions, Lyme disease (LD) has a higher incidence [3,4]. Tick-borne zoonoses have a high magnitude in North America [5,6], Europe (several countries), and Asia [7]. In addition, climate change can alter the entire ecosystem, influencing the increase in the population of ticks and their geographic distribution [8,9].

Brazil is typically a tropical region with a very diverse biome and fauna, which favors the diversity of tick species. In this scenario, the capybara, the largest rodent in the world, is considered the amplifying host of rickettsiae [10]. The ticks *A. nitens* and *A. cajennense* are ectoparasites of horses, and *A. cajennense* has a preponderant role in transmitting rickettsiae to humans in Brazil. A study conducted in the south of the country detected anti-*B. burgdorferi* in 4/87 (4.6%) humans, 26/83 (31.3%) dogs and 7/18 (38.9%) horses by RIFI [11]. Another study demonstrated the existence of different activity peaks for the larval (April to July), nymph (August to September), and adult (September to March) stages in horses throughout the year in Brazil [12].

In Figure 1, the tick life cycle is schematically presented, associated with seasons and climate variation in the natural environment, showing wild and domestic animals as maintainers of the tick population, such as capybara, horse, dog, and man as an accidental host [13].

Developing and emerging countries are expected to have higher densities of ticks, which predisposes to a more significant occurrence of tick-associated diseases. This association is evident not only in some regions of Brazil but also in other countries such as China [14], India [15], and Russia [16], but it is not the predominant factor since several developed countries also present an increase of incidence [1]. Instead, this may be explained by social reasons (modifications in human behavior, duration and type of leisure activities, increased tourism in high-risk areas) and ecological factors (e.g., effects of climate change on the tick population and reservoir animals).

Currently, Brazil has a population of 213 million, data estimated by the government’s statistics agency IBGE, of which 13.151 million live in inadequate housing. Environmental change and socioeconomic factors may be associated with TBD transmission [17]. Additionally, the dog is the most popular pet animal. In Brazil, about 46.1% of households had at least one dog in 2019 and many are known to be sentinels [18,19]. In a recent study of the occurrence of *Ehrlichia* spp. in *Xenarthra mammals* from states in Brazil (São Paulo, Mato Grosso do Sul, Rondônia, and Pará), 24.54% (81/330) were positive in PCR screening assays [20]. In another study, the significant prevalence of *E. canis* canine in Seropedica, Rio de Janeiro, was 31.1% (n =  47/150) [21]. A serological analysis conducted in 2013–2015 demonstrated that 8 to 11 % of domestic dogs were seropositive for *Rickettsia rickettsii*, 9 to 37 % for *R. amblyommatis*, and 61 to 75 % for *E. canis*, and 0–5% for *Coxiella burnetii* [13]. In 2013, 69.4% (n = 108) of dogs were seropositive to *E. canis* [22]. These last two investigations were reported in northeastern Brazil.

Another recent study using genotypic mapping of tandem repeat proteins demonstrated the existence of a wide distribution of *E. canis* genotypes in Brazil. The most prevalent are the American and Brazilian genotypes [23].

On the other hand, the presence of different pathogens (*E. canis*, *A. platys*, *B. vogeli*, *H. canis,* and several *Rickettsia* of the spotted fever group) transmitted by ticks (to horses, dogs, and men) in the State of Espirito Santo were demonstrated by IFA and/or real-time PCR [24,25].

Among the most prevalent TBDs, the only one that has a record for humans is Brazilian Spotted Fever (BSF), caused by *R. rickettsii* and *R. parkeri* strain Atlantic Forest [25,26]. According to Valente et al., there is a great diversity of ticks and hosts in Brazil, which reinforces the need for greater epidemiological monitoring of ticks in the country, mainly due to the increase in the number of reports of spotted fever [27]. Figure 2 shows the evolution of spotted fever notifications in Brazil.

Even though Brazilian Borrelia disease (BBD) is not notified, many cases have increased. The prevalence of IgG antibodies to *Borrelia* was 3.5% (16 samples) [28]. Bonoldi et al. showed that BBD patients with clinical symptoms presented 50% (14/28) positive serology in the acute phase (<3 months) and 45% (10/22) positive serology in the later phase (>3 months) [29]. In addition, ticks can transmit more than one pathogen. Multiple infections can generate similar signs and symptoms, but different diseases [30,31]. Tick diseases can demonstrate the asymptomatic subclinical phase in humans and dogs [32]. In symptomatic dogs, clinical signs (fever, pale mucous membranes, apathy, anorexia, lymph node enlargement, and weight loss) are described for canine ehrlichiosis and BSF. However, most canines with BBD are asymptomatic [33,34], but humans have acute symptoms, which should be dissociated from flu-like symptoms (fever, headache, vomiting, muscle aches, and pain) [35,36].

The greatest challenge to clinicians is not therapy but the difficult diagnosis during the early phase of infections [37]. The diagnosis of TBD may be masked due to these initial non-specific clinical presentations and the absence of confirmation by specific laboratory testing [38]. In addition, serological diagnosis is usually retrospective; antibody increase takes 15–26 days, thus limiting the clinical impact of diagnosis [39].

Although molecular tests already exist to diagnose some TBDs, the similarity of signs and symptoms with a non-absolute specificity of the molecular methods [40] leads to the need for more precise identification of pathogens in humans and animals. In this way, refined immunological diagnostic tests, which are cheaper, reassume relevant importance in controlling outbreaks before they become epidemic [41,42] and directing the best medical treatment.

Microscopy, serological and molecular methods have been applied to detect pathogen TBDs in Brazil [43,44]. The criteria recommended by the CDC and considered the gold standard are two-layer diagnostic tests, consisting of an immunofluorescence assay (IFA) and enzyme linked-immunoassay (ELISA), which, when positive, must be confirmed by a second-step immunoblot [43,45]. The diagnostic test for BSF and ehrlichiosis recommended in Brazil is a dual test for LD [42,46], even though the condition has different immunological and epidemiological aspects than LD. Thus, the analysis of the second stage of the test, western blot, needs a certain number of bands present to be considered positive [38,44]. However, even following this strict criterion, a larger study conducted in Brazil demonstrated a significant number of false positives (16%) [46].

So new immunological methods have been developed regarding immune response. When infection with a pathogen occurs, innate and acquired immunity play a role in establishing and eradicating it [47,48]. The IgM antibodies appear in the first week of infection in the humoral immune response, and IgG is second [49,50]. The quantity and affinity of these antibodies influence the tests. Immunoassays detect antibodies or antigens from pathogens, presenting the capacity to show if the patient has an ongoing infection. The applicability of these immunodiagnostic tools helps determine immune reactivity antibody-antigen.

Many immunological assays present performance that varies widely, such as ELISA, IFA, western blot (WB), and hemagglutination used in BSF [49,50,51], Lyme disease [43], and ehrlichiosis [50] diagnostic. A short overview of the methodology applied for those diseases is shown in Table 1.

In recent years, biosensor devices have attracted scientists’ attention. The biosensor uses a target analyte attached to the transducer to generate an output signal [52]. Its integration with antibodies or antigens has contributed to the development of the immunosensor. This paper aimed to review some immunodiagnostic assays used to detect TBDs, specifically borreliosis, ehrlichiosis, and BSF disease (Table 2).

**Table 1 pathogens-11-00870-t001:** Diagnosis test for tick-borne diseases.

Disease	Pathogen	Test	Method	% Sens	% Sp	% PS	Country	Ref.
		VlsE1/pepC10	ELISA IgM/IgG	59.5–80.9	86.9–93.0		USA	[53]
LD	*B. burgdorferi*	MarBlot^®^	WB IgM/IgG			84.7/87.3	USA	[54]
		VIDAS^®^Lyme	EIA IgM/IgG	83–85	85–88		USA	[55]
ATBF	*R. africae*	Focus Diagnostics	IFA IgM/IgG		95		USA	[56]
RMSF	*R. rickettsii*	Tulip Diagnostics	WF IgM/IgG	49	96		India	[57]
MSF	*R.conorii*	Vircell	ELISA IgM/IgG	94/85	95/100		Spain	[58]
	*E. chaffeensis*	Fuller Laboratories	IFA IgM/IgG			1.8/7.0	USA	[59]
HME	*E. canis*	Fuller Laboratories	IFA IgG			19	USA	[60]
	*E. canis*	ImmunoComb	ELISA IgG			4.33	Spain	[17]

PS, positive sample; Sp, specificity; Sens, sensitivity; ELISA, enzyme-linked immunosorbent assay; WFT, Weil-Felix test; IFA, Indirect fluorescent antibody assay; MSF, Mediterranean spotted fever; LD, Lyme disease; ATBF, African tick bite fever; RMSF, Rocky Mountain spotted fever; HME, Human Monocytotropic Ehrlichiosis.

**Table 2 pathogens-11-00870-t002:** Serology-based methods with advantages and disadvantages used for Lyme, ehrlichiosis, and rickettsiosis diseases.

Serology-Based Methods	Disease	Sample	Advantages	Disadvantages	Reference
ELISA	ehrlichiosis/LD	100 mL	↑ specificity	↓ sensitivity	[61]
Immunoblotting	borreliosis (LD)	0.5 mL	↑ specificity	heterogeneity, ↓ sensitivity	[62]
IFA	rickettsiosis/ehrlichiosis	25 µL	↑ sensitivity	subjective	[45]
WFT	rickettsiosis	0.1 mL		↓ sensitivity/specificity	[63]
Electrochemical	rickettsiosis	20 µL	↑ sensitivity/specificity, fast response		[64]

ELISA, enzyme-linked immunosorbent assay; LD, Lyme disease; WFT, Weil-Felix test; IFA, Indirect fluorescent antibody assay; ↑ High; ↓ Low.

### 1.1. Humoral Immunity

When a bacterial infection occurs, during the innate immune response cells recognize highly conserved pathogen structures, known as pathogen-associated molecular patterns, through pattern recognition receptors, which are expressed on the cell surface or secreted in body fluids [65,66].

B cells produce different isotypes of immunoglobulins, but the main subclasses of diagnostic importance have been IgM (early phase) and IgG (later phase). However, IgA has recently shown considerable diagnostic value for tick-borne spotted fever rickettsial in the early phase [16,60].

Figure 3 shows the immune response kinetics for Rickettsia, Ehrlichia, and Borrelia diseases. However, temporal variations of Ig subclasses have been described.

IgM is evident within the first three weeks of all infections [67]. However, in both LD and RMSF, the IgM level declines after the first week of illness, whereas ehrlichiosis remains high [66]. In addition, *Ehrlichia* and *Borrelia* can suppress or delay the onset of a germinal center’s response [68], which is vital in generating high-affinity antibodies [69].

IgG begins to be produced within 3–6 weeks of BBD, although IgM and IgG may persist for months and years [65,70]. On the other hand, in *Rickettsia* and *Ehrlichia* infections, IgG increases at around the second week and can continue for many years [65,67].

#### 1.1.1. Agglutination Tests

First described in 1916, the Weil-Felix reaction (WFT) is a test used to diagnose rickettsial infections. This test is based on specific serotypes of Proteus bacteria displaying antigenic cross-reactivity with *Rickettsia* species. Although at the same time, it has largely been replaced with new serological techniques, the WFT remains important in resource-limited areas where more advanced methods are unavailable [71]. However, due to its low sensitivity and specificity, the WFT has fallen out of favor in most clinical settings, and its use is no longer recommended in routine practice.

An indirect hemagglutination assay for immunodiagnosis of Rocky Mountain spotted fever (RMSF), using an erythrocyte-sensitizing substance from *R. rickettsii* adsorbed to latex particles, has been noticed. The test was evaluated with 123 single and 118 paired human sera submitted for RMSF testing. Its efficiency, relative to the reference micro-immunofluorescence test, was 95.1% for single sera and approached 100% for paired sera [72]. Furthermore, another latex agglutination assay was designed using a sonicate of *Borrelia microtti* flagella and showed 98% sensitivity and 95% specificity [73]. In addition, a latex agglutination capture antibodies test for *R. conorii* was compared with the micro-immunofluorescence, showing the same sensitivity and specificity [74]. However, both demonstrated low specificity and sensitivity (33%) in the acute phase of rickettsial infections [75].

The agglutination tests can form clumps of the particular material, and antibodies from patient sera can be captured by the antigen and detected in the pellet [61]. A variety of agglutination tests have been developed over the last four decades. Although simple and easy to create, they lack specificity, sensitivity, and reproducibility, which prevents their development on a large scale.

#### 1.1.2. Immunoblotting (Western Blotting)

Western blotting (WB) is a semi-quantitative or quantitative valuable method for identifying antigenic protein bands and is sometimes also employed in diagnosis to recognize specific antigens [76,77]. Among the TBD infections mentioned previously, LD is the only one yet to perform this type of assay for confirmatory diagnosis. In Brazil, it has been used to clarify ambiguous results of the first molecular analysis and uses the diagnostic criteria recommended and adopted by the CDC/U.S.A.

A study also conducted in Brazil to confirm the acute phase of the disease concluded that the test shows low specificity. Of 82 patients’ sera screened for IgM-Lyme, 50 were false positives (27.5%, 95% CI: 21.1–34.6) [78]. In another study, in 212 cases evaluated, 113 (53.3%) were false positives for IgM [35]. Likewise, a meta-analysis study comparing different assays detected heterogeneity, low sensitivity, and high specificity, which may be related to the type of sample, stage of disease, and subjectivity in interpreting results [79].

#### 1.1.3. Indirect Immunofluorescent Assay

The indirect immunofluorescence assay (IFA) is a simple test with low specificity and high rates of false-negative results [80]. Likewise, extensive cross-reactions between *E. canis* and *Ehrlichia* spp. have been described [81,82]. Despite these contradictions, such as lack of sensitivity, imprecision, and time consumption, a fluorescence microscope is needed, which prevents it from being available in many endemic regions. Today, the IFA is the serological diagnostic method is considered the gold standard to confirm *Rickettsia* and *Ehrlichia* infection [44,75]. The test requires two paired serum samples taken within 2–3 weeks post-infection, and the result is considered positive if the antibody titer is ≥64 [76]. However, it should not be used to diagnose the acute phase (IgM) because, in the first two weeks of infection, it has low sensitivity and cross-reactions to various bacterial antigens [83,84,85].

#### 1.1.4. Enzyme-Linked Immunosorbent Assay (ELISA)

The ELISA is one of the most common assays used under different formats for diagnosis. This selection is simple and efficient and can be developed in two or three steps (capture of antibodies or circulant antigen) [51]. However, to increase the sensitivity and specificity of the assays, it is necessary to identify previously robust and specific antigens, which has been a challenge up to now.

Recombinant proteins [69,83] and peptides selected by phage display [86] have usually been employed to diagnose TBD, but the results were generally not encouraging. One study using peptide-based ELISAs to diagnose anaplasmosis and Ehrlichiosis in dogs obtained higher sensitivity but lacked specificity [87]. Another work analyzed three ELISA-recombinant proteins for detecting anti-*Ehrlichia canis* IgG antibodies, but the results demonstrated a low sensitivity [87]. A third ELISA has been licensed by the FDA to see anti-IgM *R. typhi*, but has a sensitivity of 45% and a specificity of 98.3% [88].

#### 1.1.5. Immunosensors

Electrochemical sensors are powerful tools in analytical chemistry. They are composed of electrical transducers that measure the environment’s chemical changes, mainly potentiometric, amperometric, and conductometric. Commercial electrochemical sensors lead the market, and they are applied in essential fields of industrial, environmental, agriculture, and clinical analysis [71]. For the latter, an annual growth of 7.4% is expected for the 2021–2027 period [86,89]. Moreover, this tool catches the eye for its remarkable detectability, experimental simplicity, low cost, fast response, production scalability in the case of screen-printed technology, and it is conducive to miniaturization [89]. Furthermore, the emerging two-dimensional layered materials incorporated into biosensor devices have improved sensitivity [90,91].

Our group developed a simple and elegant electrochemical biosensor using a commercial screen-printed electrode for Spotted fever diagnosis [64]. The working electrode was modified with glutaraldehyde to allow epitopes of peptides’ covalent bond followed for the specific and sensitive detection of IgG. Based on detailed microarray analysis of outer membrane protein-A of *R. rickettsia*, an epitope available to the immune system and recognizable by B cells was chosen to serve as a binding site for IgG. In this way, concerns related to the immunofluorescence golden standard test such as lack of sensitivity at the acute phase, requirements for the analysis of paired sera collected over 2–3 weeks, and the highly controlled environment to establish the quality of the antigen. Furthermore, a low-cost, portable, and faster detection than ELISA, which requires a small volume of samples, was developed using sec-IgG labeled with alkaline phosphatase.

Recently, another dispositive that was developed was the bioconjugation of silver nanoparticles (AgNPs) with the sec-IgG for signaling markers to quantify IgG anti-tick-borne encephalitis (TBE) virus (TBEV) [92]. The immunosensor comprised a carbon composite electrode modified with gold nanoparticles produced electrochemically and via cysteamine and glutaraldehyde. TBEV antigen was covalently conjugated as an IgG-binding site. AgNPs, dissolved in HNO_3_ solution, recognized the captured IgGs, and the resulting free Ag ions were quantified successfully by cathodic linear sweep. Thus, they produced a cheaper and more stable marker than others based on enzymes.

More recently, another rapid, cheap, and simple detection method of TBEV using cyclic voltammetry (CV) was established [93]. A supervised neural network model CV response of the eutectic gallium indium alloy/agar hydrogel modified with TBE antibody was fed to detect the virus indirectly. The antigen/antibody interaction was detected by changing the gel composition, and the supervised method identified the CV response patterns. The work showed proof-of-concept results and demonstrated a high potential to develop machine learning detection.

### 1.2. Immune Disorder and Diagnostic Implication

Great variations are observed in the antibody response to *B. burgdorferi* infections. Some patients maintain a detectable antibody titer for years, for others titers decline over time, and some never present antibodies [65]. This phenomenon has also been observed in animals, leading to a hypothesis of a general dysregulation of adaptive immunity [37]. Ultimately, the antibody response mounted by infected individuals is mostly ineffective in completely eradicating spirochetes and/or establishing long-term immunity [94,95,96]. A *B. burgdorferi* infection can redirect the adaptive immune system from a long-term protective antibody response toward a less efficacious short-lived antibody response that can be both rapid and strong [97].

Throughout a *B. burgdorefi* infection, multiple changes in lymph nodes have been observed that can be divided into four phases. Initially, B cells accumulate in lymph nodes, inducing antibody production independent of T-cells. The typical architecture is altered by changes in the organization of B cell follicles and T cell zones [98]. The appearance of spirochetes between 5–10 days post-infection further varies B cell follicles [99]. In addition, B cells began accumulating in large numbers that can reach greater than 70%, which skews its ratio to T cells [100]. The germinal centers’ formation around 2–3 weeks later marks the second phase. Short-lived centers generate low numbers of antibody-producing plasma cells within the bone marrow. The third phase covers the slow accumulation of plasma cells. In the final phase, lymph node germinal centers begin to disappear around week four, even in the presence of bacteria. The timing of B cell accumulation before the changes in lymph node morphology suggests that the appearance of bacteria is responsible for the atrophy of lymph nodes [101].

The CDC recommends a two-tiered testing program for suspected cases of LD to detect *B. burgdorferi*-specific antibodies and an algorithm to diagnose an infection [100]. The first-tier test is an ELISA. Positive and borderline results are retested in the second tier, which consists of a Western Blot on bacterial proteins. In theory, both IgM and IgG antibodies are evaluated, and this test can identify active infections [101]. However, this approach may generate misleading diagnoses due to the duration of the IgM response. In a cohort of individuals with resolved LD, 13% (10 of 79) presented IgM antibody in the two-tiered test 10–20 years after the infection [102]. Nearly half (34 of 79) continued to display IgG reactivity. In shorter time intervals between testing and the disease, these percentages were slightly higher; 15% (6 of 39) with IgM and 62% (24 of 39) with IgG. Similar results have been observed in mice, where IgM antibodies do not decrease even as the IgG levels rise [99]. The persistence of IgM, even after recovery, could interfere with healthcare decisions and result from the described alterations in lymph node architecture [99,100]. Additional studies are needed to understand the IgM response’s duration and the absence of long-lived plasma and memory cells.

For LD diagnosis, it is important to differentiate between active and inactive infections in a format that can dispense with the current two-tier approach. Therefore, an active area of research is to focus on identifying antigens that are expressed very early in LD (e.g., VlsE1 and pepC10) [101,102], which have enhanced the performance of the diagnostic assays. Additional areas include nucleic acid and antigen detection [100]. In this regard, a multiplexed test employing three antigens (VlsE, PepVF, and OspC) combined with a microfluidic chip technology platform (mChip-Ld) showed high sensitivity and efficacy for use in the identification of the early stage of LD [103]. Likewise, surface-amplified Raman spectroscopy was used in combination with aptamers to identify the OspA protein (present during active LD infection) in human serum at an extremely low concentration (10–4 ng OspA/mL serum) [104].

PCR assays have been developed to detect *B. burgdorferi* DNA [105]. Directly seeing spirochetal components can be an accurate method to identify active infections, such as antigens for OspC [105] and peptidoglycan [106]. OspC is present on spirochetes originating from the tick to the mammalian host [103]. It can detect peptidoglycan in patients after antibiotic treatment and is the cure for active infection [107]. Since metabolically active spirochetes only produce peptidoglycan, there is a possibility that a *B. burgdorferi* could transition to a persistent infection.

## 2. Conclusions

Brazil is the most populated country in Latin America, with high diversity in ecosystems. Therefore, environmental change and animal locomotion contribute to an increased tick population. In addition, the number of dogs in households has increased in recent decades. Consequently, the risk of diseases transmitted by ticks has also increased. However, BSF is the only one of the TBDs with mandatory notification diseases, which directly influences the questioning of the clinical diagnosis of BSF, ehrlichiosis, and BBD. In the initial phase, the clinical symptoms are flu-like, and the immunologic method demonstrates variation in both specificity and sensitivity, especially low specificity resulting in cross-reaction. These limitations indicate an urgent need to find and validate direct molecular markers derived from the pathogen to improve new, more specific, and fast immunologic methods. Strategies for these methods should also encompass high sensitivity and specificity, simple and intuitive handling, low sample volume, and quick results. In addition, it is necessary to better understand the mechanisms of dysregulated antibody response described for Lyme disease patients since this is critical in determining the sensitivity for developing improved diagnostic tests for this TBD [42].

Tests using immunosensors are being presented as promising for detecting TBDs at an early stage with high sensitivity and precision. However, they still need to be available in commercial devices to be applied on a large scale. Therefore, different tests using ELISA or immunochromatography are the most practical to be developed. In this respect, recombinant proteins carrying multiple specific epitopes [108] are a diagnostic strategy that can improve the specificity and sensitivity required for tick-borne diseases diagnosis and may also contribute to the epidemiology of *E. canis E. chaffeensis* and *B. burgdorferi*. Finally, this may progress in diagnosing and treating borreliosis, Ehrlichiosis, and BSF to monitor animal and human health.

## Figures and Tables

**Figure 1 pathogens-11-00870-f001:**
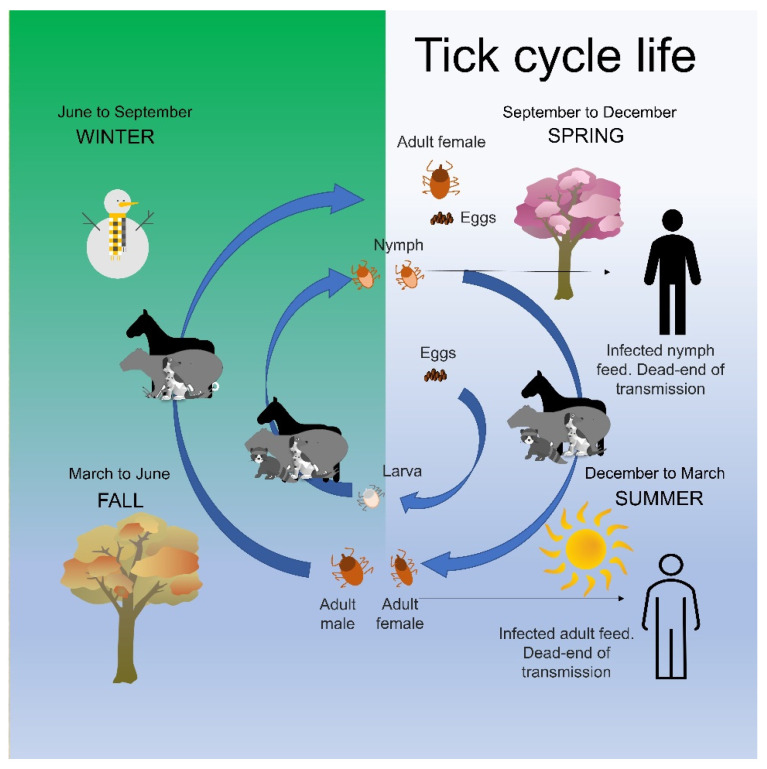
Tick disease transmission life cycle (*Rickettsia*, *Ehrlichia*, *Borrelia*, and others) in Brazil. Horses and capybara are part of the tick cycle in Brazil. However, it is essential to understand that each tick may be associated with transmitting more than one pathogen.

**Figure 2 pathogens-11-00870-f002:**
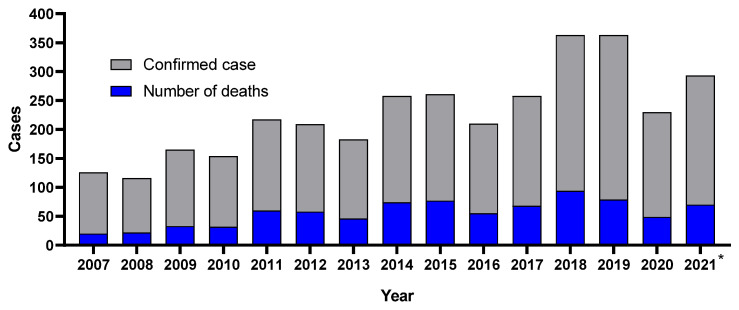
Number of cases of Brazilian Spotted Fever in the period 2007–2021 in Brazil [23]. * *p* < 0.05.

**Figure 3 pathogens-11-00870-f003:**
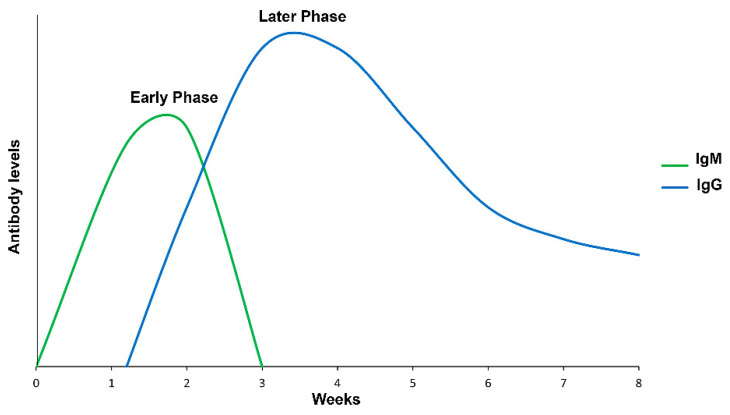
Evidence of the immune response to rickettsiosis, ehrlichiosis, and borreliosis. The IgM response can be detected in the first days with a major peak in the first three weeks, while IgG is seen between the first and second week and can persist for a long time, depending on the disease.

## Data Availability

The data presented in this study are available on request from the corresponding author.

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
