# Peer review of "Recent Advances in the Immunologic Method Applied to Tick-Borne Diseases in Brazil"

_pathogens, 2022, doi:10.3390/pathogens11080870_

Round 1

Reviewer 1 Report

Overall, the article is interested and the topic is relevant. However, it needs English editing. The proper terms were not used in parts and it lost sense in others. 

Figure 1. Large animals, like horses, are not the normal host for tick larvae. Please consider editing the figure to add small mammals instead.

Line 107 - This is the recommendation for Lyme disease. Not all TBDs, please clarify. 

Line 158 refers to figure 2. 

Line 298 and after, B. burgdoferi should be in italics.

This is a review. It does not require a material and methods section.

Other diagnostic advances need to be discussed: 

An antigen-targeting assay for Lyme disease: Combining aptamers and SERS to detect the OspA protein

Biomarker selection and a prospective metabolite-based machine learning diagnostic for lyme disease      

Author Response

Overall, the article is interesting and the topic is relevant. However, it needs English editing. The proper terms were not used in some parts and lost meaning in others.

1)Figure 1. Large animals such as horses are not the normal hosts for tick larvae. Please consider editing the figure to add small mammals.

R: Thank you. Small mammal hosts have been inserted in the figure,  however, in Brazil, the horse and capybara are hosts of the tick's biological cycle and are part of the Health Surveillance Manual of the Ministry of Health of Brazil, therefore they were maintained in the figure.

2) Line 107 - This is the recommendation for Lyme disease. Not all TBDs, please clarify.

R: The text has been modified for better clarification to the reader. “The criteria recommended by CDC considered gold-standard are indirect immunofluorescence assay, requiring paired samples of sera collected during the acute and convalescent phase, for the diagnosis of BSF and Ehrlichiosis, and for Brazilian Lyme-like disease is necessary indirect immunofluorescence assay or enzyme immunoassay (ELISA) that, when positive, should be performed by a second step immunoblot [43,45].”

3) Line 158 refers to figure 2.

R: Thank you, this has been corrected.

4) Line 298 and after, B. burgdoferi should be in italics.

R: Thank you, fixed.

5) This is a review. It does not require a materials and methods section.

R: We totally agree and this section has been removed.

6) Other diagnostic advances need to be discussed: An antigen targeting assay for Lyme disease: Combining aptamers and SERS to detect OspA protein

7) Biomarker selection and a prospective metabolite-based machine learning diagnosis for Lyme disease

R: Thank you, all suggestions have been included.

Other approaches, for early-stage identification, were employed by the single multiplexed test with 3 VlsE, PepVF and OspC antigens combined with the mChip-Ld platform microfluidic chip technology resulting in effective and high sensitivity [105]. Surface-amplified Raman Spectroscopy combined with aptamers was shown to identify OspA protein (present during active Lyme infection) in human serum at a lower concentration (1 × 10-4 ng OspA/mL serum) and obtained a 50% increase in serum sensitivity when compared with current tests for Lyme [106]. The search for new targets and methods for the development of diagnosis has been advancing more and more in obtaining a quick test, low sample volume and more sensitive. Thus, other methods have been developed, such as the combination of metabolomics technique and machine learning to identify markers, Kehoe and collaborators obtained a profile of metabolic biomarkers of seropositive for Lyme. Sera samples were analyzed by chromatography-mass spectrometry (LCMS) and these data were used to create an artificial intelligence method with the aim of differentiating between metabolite patterns of Lyme seropositive and serum from uninfected individuals. According to the data obtained, they had a 98.13% success rate, 96.25% sensitivity, and 100% specificity. The method has demonstrated good performance for the metabolite-based diagnostic test for Lyme disease [107].

Reviewer 2 Report

This manuscript is a review of available diagnostic tests for tick-borne bacterial diseases. 

I suggest that instead of "immunological methods" you change the wording to "serological methods"

Overall, the paper provides some information on current diagnostic tests, and introduces a few new ones.  However, for the newer tests, no data directly comparing specificity and sensitivity to that of other assays is provided.  Nor is there sufficient discussion of how these new assays may overcome the shortcomings of previously developed assays.  Do the new assays require purchase of special equipment? What is the cost of that? 

The introduction is too broad, focusing on all tick borne diseases, and does not provide much specific information on bacterial tick borne diseases of concern in Brazil. 

The authors state (line 19) that 75% of 650,000 cases of infectious disease in the U.S. during a given period of time were tick borne disease.  I read the referenced article, and cannot find where that statistic is stated (and think it is inaccurate).  Please address.  

In line 36, change "the vectors" to "arthropod vectors"

Line 93 is irrelevant -- although zoonotic, COVID is not a tick borne disease. Please remove.

In line 297, change "Disfunction" to "Dysfunction"

In general, there are numerous grammatical and English syntax errors -- extensive editing is required.  

Author Response

This manuscript is a review of available diagnostic tests for tick-borne bacterial diseases.

1) I suggest that instead of "immunological methods" you change the wording to "serological methods"

R: Thanks, this has been corrected.

2) Overall, the article provides some information about current diagnostic tests and introduces some new ones. However, for the more recent tests, data that directly compare specificity and sensitivity with other assays are not provided. Nor is there sufficient discussion of how these new assays might overcome the shortcomings of previously developed assays. Do new trials require the purchase of special equipment? What is the cost of this?

R: For clarity, a new table has been added (Table 1) indicating the effectiveness and sensitivity of commercial tests used in various countries. In addition, we have included a discussion of how the new trials can overcome the shortcomings of the previous ones.

3) The introduction is very broad, focusing on all tick-borne diseases, and does not provide much specific information on bacterial tick-borne diseases of interest in Brazil.

R: Thank you, the text has been modified as suggested to provide more information about tick-borne diseases in Brazil.

“Brazil is a typically tropical region with a very diverse biome and fauna, which favors the diversity of tick species and in this scenario the capybara, the largest rodent in the world, is considered the amplifying host of rickettsiae [10]. The ticks A. nitens and A. cajennense are ectoparasites of horses and A. cajennense has a main role in the transmission of rickettsiae to humans (Campos). In southern Brazil, anti-B. burgdorferi were detected in 4/87 (4.6%) humans, 26/83 (31.3%) dogs and 7/18 (38.9%) horses by RIFI [11]. Another study demonstrated different activity peaks for the larvae (April to July), nymph (August to September) and adult (September to March) stages in horses throughout the year in Brazil [12] a figure 1, the tick life cycle is schematically presented, which is associated with seasons and climate variation in the natural environment presenting wild and domestic animals as maintainers of the tick population, such as capybara, horse, dog and man as an accidental host [13”]

In 2020, the study of genotypes of tandem repeat proteins (TRPs), such as American (USTRP36), Brazilian (BrTRP36) and Costa Rican (CRTRP36) demonstrate a wide distribution of E.canis in Brazil, however the American genotypes and Brazilians of E. canis were more prevalent and evenly distributed in Brazil [24]. and currently Rickettsia rickettsii and Rickettsia parkericepa strain Atlantic Forest are recognized as human pathogens [27,28]. According to Valente et.al., there is a great diversity of ticks and hosts in Brazil, which reinforces the need for epidemiological monitoring of ticks in the country, mainly due to the increase in the number of notifications[28], shown in Figure 2.

In symptomatic dogs with ehrlichiosis present clinical manifestations divided into three phases: acute, asymptomatic (subclinical) and chronic, clinical signs in the acute phase can be (fever, pale mucous membranes, apathy, anorexia, lymph node enlargement, and weight loss) and can manifest in both

4) The authors state (line 19) that 75% of the 650,000 infectious disease cases in the US during a given time period were tick-borne diseases. I read the referenced article and I can't find where this statistic is stated (and I think it's inaccurate). Please ENTER THE REFERENCE.

R: Although this information is in the abstract, we have included the reference.

5) On line 36, change "the vectors" to "arthropod vectors"

R: Thanks, modified

6) Line 93 is irrelevant - although zoonotic, COVID is not a tick-borne disease. Please remove.

R: Thanks, it was deleted

7) On line 297, change "Disfunction" to "Disfunction"

R: Please, this has been corrected.

8) In general, there are several grammatical and syntax errors in English – extensive editing is required.

R: The manuscript was reviewed by a native researcher.

Reviewer 3 Report

Although your manuscript could be a good review, this current version can not be accepted how it is presented. An extensive correction-edition is required before submitted again. Please consider an English edition, and be careful when scientific names are writing. 

Regards

Author Response

We appreciate the referee's effort and suggestions. All the suggestions pointed out in the pdf were included in the new version. The manuscript was reviewed by a native researcher.

Round 2

Reviewer 2 Report

Thank you for your revisions.

Regarding your assertion that 75% of 650,000 infectious disease cases were tick-born (abstract line 18-19):  I have looked through the referenced website given, and cannot see where on the website you found this statistic.  How did you get to a total of 650,000 cases of infectious disease?  The U.S. sees MANY more infectious diseases than this per year, so this makes very little sense.  This needs corrected, as it is inaccurate.  

The English syntax is much improved from the last version, but please do additional checks for small mistakes.

Author Response

Thank you for your revisions.

Regarding your assertion that 75% of 650,000 infectious disease cases were tick-born (abstract line 18-19):  I have looked through the referenced website given, and cannot see where on the website you found this statistic.  How did you get to a total of 650,000 cases of infectious disease?  The U.S. sees MANY more infectious diseases than this per year, so this makes very little sense.  This needs corrected, as it is inaccurate.  

The English syntax is much improved from the last version, but please do additional checks for small mistakes

R: We appreciate the referee's effort and time. As the information in this paragraph was confusing, we removed it from both the abstract and the introduction, and a new reference was inserted (in color). In addition, a new revision, including correction software, was carried out regarding grammatical and syntax errors. We believe that the manuscript is now ready to be accepted for publication.

Reviewer 3 Report

Dear authors, 

I would appreciate a cover letter addressing all my previous observations and comments in order to proceed with a second review. 

Please note that citations are not allowed in the abstract. 

I have look at the document briefly and detected several mistakes such as: ---Ehrlichetiose, 

-Ehrlichiosis, Lyme, and Rickettsiosis OR Ehrlichia, BSF, and Borreliosis???

-(Richectssia, Ehrlichia, Borrelia, and others)

-The document is presented with ortographic errors and gramatical mistakes.

-A deep edition of both, technical and English is required. 

Author Response

Comments Referee 3

1st Round: An extensive correction edition is required before being submitted again. Please consider an English edition, and be careful when scientific names are written.

R:  We appreciate the effort and time spent by the referee and the pdf forwarded with various syntax and typo corrections. We agree with the reviewer’s assessment. Therefore, throughout the manuscript, we have revised including using a word and sentence correction software.

2nd Round
Comment 1: Please note that citations are not allowed in the abstract
R: Thank you for pointing this out. Were inserted the references in the abstract because reviewer one had questioned the lack of citation. As we took this information from both the abstract and the introduction, we also suppressed the references from the text in this version.Comment 2: I have a look at the document briefly and detected several mistakes such as ---Ehrlichetiose, Ehrlichiosis, Lyme, and Rickettsiosis OR Ehrlichia, BSF, and Borreliosis ??? -(Richectssia, Ehrlichia, Borrelia, and others)
R: Thank you for pointing this out. We also use word correction software to correct these errors throughout the text and to minimize any syntax or typing errors as much as possible.

Comment 3: The document is presented with orthographic errors and grammatical mistakes. A deep edition of both technical and English is required.
R: Thank you for pointing this out. We are in according that the text was confusing. The words were now uniformized and abbreviated to minimize the different ways in the world to express a given disease. Also corrected any syntax or typing errors.